

# Ubiquitin-proteasome system in diabetic retinopathy

Zane Svikle[1], Beate Peterfelde[1,2], Nikolajs Sjakste[1], Kristine Baumane[1,2], Rasa Verkauskiene[3], Chi-Juei Jeng[4,5] and Jelizaveta Sokolovska[1]

[1] Faculty of Medicine, University of Latvia, Riga, Latvia
[2] Ophthalmology Department, Riga East University Hospital, Riga, Latvia
[3] Institute of Endocrinology, Lithuanian University of Health Sciences, Kaunas, Lithuania
[4] Ophthalmology Department, Taipei Medical University Shuang Ho Hospital, Ministry of Health and Welfare, Taipei, The Republic of China (Taiwan)
[5] College of Medicine, Graduate Institute of Clinical Medicine, National Taiwan University, Taipei, Taiwan

## ABSTRACT

Diabetic retinopathy (DR) is the most common complication of diabetes, being the most prevalent reason for blindness among the working-age population in the developed world. Despite constant improvement of understanding of the pathogenesis of DR, identification of novel biomarkers of DR is needed for improvement of patient risk stratification and development of novel prevention and therapeutic approaches. The ubiquitin-proteasome system (UPS) is the primary protein quality control system responsible for recognizing and degrading of damaged proteins. This review aims to summarize literature data on modifications of UPS in diabetes and DR. First, we briefly review the structure and functions of UPS in physiological conditions. We then describe how UPS is involved in the development and progression of diabetes and touch upon the association of UPS genetic factors with diabetes and its complications. Further, we focused on the effect of diabetes-induced hyperglycemia, oxidative stress and hypoxia on UPS functioning, with examples of studies on DR. In other sections, we discussed the association of several other mechanisms of DR (endoplasmic reticulum stress, neurodegeneration *etc*) with UPS modifications. Finally, UPS-affecting drugs and remedies are reviewed. This review highlights UPS as a promising target for the development of therapies for DR prevention and treatment and identifies gaps in existing knowledge and possible future study directions.

# INTRODUCTION

The number of patients with diabetes mellitus is increasing steadily throughout the world. Diabetes is a chronic condition characterized by hyperglycemia. In the case of type 1 diabetes (T1D), which is an autoimmune disease, hyperglycemia results from the autoimmune destruction of pancreatic beta cells. In type 2 diabetes (T2D) hyperglycemia develops due to pancreatic beta-cell dysfunction and insulin resistance (*Shruthi et al., 2019*). Diabetes is associated with increased morbidity and mortality mainly due to the development of neurovascular complications. Diabetic retinopathy (DR) is the most

Corresponding author
Jelizaveta Sokolovska,
Jelizaveta.Sokolovska@lu.lv

common complication of diabetes, being the most prevalent reason for blindness among the working-age population in the developed world (*Yau et al., 2012*; *Langelaan et al., 2007*). DR is characterized by microangiopathy (*Satofuka et al., 2009*) and neurodegeneration (*Ozawa et al., 2011a*; *Kurihara et al., 2008*). Microangiopathy (functional and structural changes of small blood vessels) is staged clinically according to the proliferative status of the retinal vasculature (*Kizawa et al., 2006*; *Tang & Kern, 2011*). Initially, retinal endothelial cell dysfunction appears with a loss of pericytes (*Cade, 2008*) and the development of capillaries with enhanced permeability and leukocyte adhesion (*Fernandes et al., 2014*) which leads to vascular obliteration, retinal ischemia and the resulting neovascularization (*Ozawa et al., 2011a*; *Fernandes et al., 2014*). DR is also known as one of the inflammatory retinal diseases, where inflammatory cytokines influence protein metabolism (*Li et al., 2009*; *Ozawa et al., 2011b*).

Despite the constant improvement of understanding of the pathogenesis of DR, identification of novel biomarkers of DR is needed for improvement of patient risk stratification and development of novel prevention and therapeutic approaches.

The ubiquitin-proteasome system (UPS) is the primary protein quality control system responsible for recognizing and degrading damaged proteins. UPS's impaired functions may be involved in progression of diabetes and its complications, including DR (*Shruthi et al., 2019*; *Goru, Kadakol & Gaikwad, 2017*; *Shruthi, Reddy & Reddy, 2017*). However, few recent literature reviews summarize the role of UPS specifically in DR. A summary of recent findings in the field is needed to structure existing data and help identify gaps in knowledge on UPS in DR.

In this review, we briefly describe the physiologic regulation of UPS and provide an overview of the data on changes in UPS regulation in diabetes and DR.

## SURVEY METHODOLOGY

The literature search was conducted in the PubMed, Medline, and Google Scholar databases. Emphasis was placed on articles published since 2015, but earlier articles were also included. The following keywords were used: proteasomes, the ubiquitin-proteasome system, retinopathy, diabetes, diabetic retinal disease, diabetic eye disease, diabetic macular oedema. We included original studies and reviews in English that contained information about UPS in diabetic complications, emphasizing diabetic eye disease. We also used articles cited in the reference lists of identified publications if appropriate.

### Functioning of the ubiquitin-proteasome system (UPS)

UPS is essential in regulating the cell cycle (progression, proliferation, apoptosis), immune response, inflammatory response, endoplasmic reticulum-associated degradation of proteins, and protein misfolding. Its deregulation leads to multiple disturbances in the normal cell functioning (*Zhang et al., 2016*; *Bard et al., 2018*).

The UPS includes ubiquitin, ubiquitinating enzymes, proteasome, substrate proteins, and deubiquitinases (DUBs). UPS-meditated protein degradation starts with ubiquitination and continues with proteasomal degradation. During the ubiquitination process, the ubiquitin proteins can be covalently coupled to a target protein by sequential

actions of ubiquitination enzymes. These ubiquitination enzymes include ubiquitin-activating enzyme (E1), ubiquitin-conjugating enzyme (E2), and ubiquitin-protein ligase (E3). Target proteins are covalently tagged with ubiquitin, a small protein with 76 amino acids (*Lim & Tan, 2007*). Initially, ubiquitin is activated by E1 in an ATP-dependent manner, and then the activated monoubiquitin molecule is transferred to a cysteine residue of the E2 enzyme (*Zhang et al., 2016*; *Shukla & Rafiq, 2019*). E2 receives ubiquitin from E1 and prepares it for conjugation (*Thibaudeau & Smith, 2019*). E3 identifies specific substrates and shifts ubiquitin from E2 to the lysine residue of a targeted protein, forming a polyubiquitin chain which transfers the intended protein to the proteasome for degradation (*Zhang et al., 2016*; *Shukla & Rafiq, 2019*; *Bai et al., 2016*). Eukaryotic cells contain more than 1,000 types of E3; different substrate proteins depend on the specific E3 (*Shukla & Rafiq, 2019*; *Qian et al., 2020*; *Nakayama & Nakayama, 2006*). The fate of ubiquitinated substrate depends on the number of added ubiquitin molecules. The polyubiquitinated substrates, with four or more ubiquitins attached, are recognized and degraded by the 26S proteasome. Monoubiquitinated substrates are not degraded however the biological activity or function of these proteins are altered, for example, monoubiquitination of FoxO4 promotes its nuclear translocation and enhances transcriptional activity (*Bai et al., 2016*). The site of polyubiquitination is also important. Ubiquitinylation linked at Lys 48 is a "canonical" signal for proteasomal degradation, ubiquitin attached *via* Lys 63 triggers mostly non-proteolytic processes like modification of substrate activities, modulation of protein localization or interactions, DNA damage repair, signal transduction, endocytosis, transcriptional regulation, and cell-cycle progression. Lys 27 ubiquitinylation is important in development of the innate immunity; Lys 29, in neurodegeneration and cell signaling; Lys 6, in autophagy and DNA damage response; Lys 29, in DNA damage response and cell cycle progression (*Tracz & Bialek, 2021*). After polyubiquitination (*e.g.*, four ubiquitins attached), substrate proteins are transferred to the 26S proteasome for breakdown (*Thibaudeau & Smith, 2019*; *Pickart, 2001*; *Ling et al., 2019*). The geometry of the ubiquitin chain also influences the proteolysis efficiency: branched ubiquitin chains bind stronger to proteasome receptors, and proteins with such chains are degraded more rapidly (*French, Koehler & Hunter, 2021*).

Proteasomes exist inside cells in multiple forms, including proteasome complexes with different regulatory particles to carry out protein degradation. Eukaryotic cells contain constitutive 20S core proteasome (*Thibaudeau & Smith, 2019*; *Homma & Fujii, 2020*; *Arendt & Hochstrasser, 1997*; *Jung, Catalgol & Grune, 2009*; *Groll & Huber, 2003*; *Korovila et al., 2017*; *Köhler et al., 2001*; *Stadtmueller & Hill, 2011*). When the 19S regulator binds the core proteasome (*Shukla & Rafiq, 2019*; *Jung, Catalgol & Grune, 2009*; *de Poot, Tian & Finley, 2017*), the 26S proteasome is formed (*Stadtmueller & Hill, 2011*; *Hershko, 2005*; *Tanaka, Mizushima & Saeki, 2012*; *Kudriaeva & Belogurov, 2019*). Six proteasome subtypes differing in the combination of catalytic subunits found in their catalytic chamber have been described in mammals: the standard proteasome, the two intermediate proteasomes, immunoproteasome, thymoproteasome, and spermatoproteasome. The above proteasome subtypes degrade ubiquitinylated proteins with equal efficiency. These proteasome subtypes differ in other functions. For instance, the immunoproteasome is

essential for an immune response *via* promoting a pro-inflammatory environment. Moreover, intermediate proteasomes and immunoproteasomes are more efficient than the standard proteasome in the ATP- and ubiquitin-independent degradation of oxidized proteins and proteins containing intrinsically disordered regions (*Chapiro et al., 2006*; *Abi Habib et al., 2022*).

Nearly 100 DUBs are expressed by the human genome to regulate the ubiquitination process (*Komander, Clague & Urbé, 2009*). DUBs can remove ubiquitin from substrates and deconstruct polyubiquitin chains, leading to protein stabilization (*Zhang et al., 2016*). DUBs usually have various substrates and are cell-specific. The interaction between ubiquitination and deubiquitination appears to regulate the equilibrium of proteasomal degradation, cell cycle progression, gene expression, apoptosis *etc.* (*Gupta et al., 2018*).

The UPS is also involved in the degradation of misfolded secretory proteins and most integral membrane proteins in the endoplasmic reticulum (ER) for proper folding through the protein quality control system-ERAD (endoplasmic reticulum-associated protein degradation) pathway (*Thibaudeau & Smith, 2019*). Proteins in the ERAD system are extracted from the ER and degraded in the cytosol or ER membrane (*Kaneko et al., 2017*). E3 ligases of ERAD ubiquitinate non-functional proteins that are collected in the ER, for the proteasomal degradation, thereby protecting against ER stress-induced cell death (*Kaneko et al., 2002*; *Kaneko & Nomura, 2003*). Unfolded protein response (UPR) activates when misfolded proteins are accumulated in the ER (*Thibaudeau & Smith, 2019*). Multiple pathologies and physiological states, like genetic mutations and oxidative stress, cause the accumulation of misfolded proteins in ER and induce UPR activation. UPR has a protective function to restore ER homeostasis, but in prolonged stress situations, UPR activation leads to ER-induced cell death (*Thibaudeau & Smith, 2019*; *Back & Kaufman, 2012*).

## UPS in the development and progression of diabetes

UPS's impaired functions may be involved in the progression of pancreatic beta cell dysfunction. For example, hyperglycemia may decrease proteasome activity, thus contributing to their ER stress, dysfunction and apoptosis (*Shruthi et al., 2019*; *Homma & Fujii, 2020*; *Back & Kaufman, 2012*; *Tang et al., 2018*; *Broca et al., 2014*). Long term activation of UPR due to hyperglycemia contributes to the development of insulin resistance as well (*Thibaudeau & Smith, 2019*; *Lenna, Han & Trojanowska, 2014*; *Hotamisligil, 2010*). Initially beta cells activate proinsulin synthesis to adapt to insulin resistance, but the increased burden of the proinsulin concentration for ER does not allow proper proinsulin folding and trafficking (*Back & Kaufman, 2012*; *Eizirik & Cnop, 2010*). ER stress triggers the UPR to remove misfolded proinsulin and to re-establish protein homeostasis. If protein misfolding persists, beta cells eventually die (*Liu et al., 2010*).

UPS is involved in the development of autoimmune diabetes (*Kaneko & Nomura, 2003*; *Liu et al., 2010*; *Thomaidou, Zaldumbide & Roep, 2018*). In the presence of insulitis, proinflammatory cytokines interrupted the homeostasis of ER, leading to ER stress (*Eizirik, Colli & Ortis, 2009*), that activated ER sensors: inositol-requiring enzyme 1α (IRE1 alpha), PRKR-like ER kinase (PERK) and ATF6, triggering the UPR (*Thomaidou,*

*Zaldumbide & Roep, 2018*; *Oyadomari et al., 2002*). UPR predisposes to activation of chaperone protein synthesis, reducing protein translation into the ER to restore ER homeostasis (*Thibaudeau & Smith, 2019*). This adaptive phase is considered to initiate the development of autoimmunity (*Thomaidou, Zaldumbide & Roep, 2018*).

Nuclear Factor KappaB (NF-kB) transcription factors regulate the expression of genes involved in inflammation, immunity, and beta-cell development. NF-kB activation is mediated through proteasomal degradation for transcriptional activation (*Thibaudeau & Smith, 2019*). Ubiquitin-editing protein A20 (tumor necrosis factor alpha-induced protein 3, TNFAIP3) acts as a negative ubiquitin-dependent regulator of NF-kB (*Catrysse et al., 2014*) and is a potent anti-inflammatory signaling molecule. There are indications of the involvement of A20 dysfunction in autoimmune and inflammatory diseases, including diabetes (*Fukaya et al., 2016*). Several mutations in A20 have been associated with T1D (*Bergholdt et al., 2012*). A20 protected mice from streptozotocin-induced diabetes (*Yu et al., 2004*) possibly impacting beta cell survival pathways (*Fukaya et al., 2016*).

## UPS-associated genetic factors, diabetes and DR

In humans, polymorphisms in *PSMA3, PSMA6*, and *PSMC6* proteasome genes are associated with T1D in a cohort of Latvian patients (*Sjakste et al., 2016*). Moreover, correlations have been revealed between some polymorphisms of proteasome genes and 42 T1D-susceptible genes encoding proteins involved in innate and adaptive immunity, antiviral response, insulin signaling, glucose-energy metabolism, and other pathways implicated in T1D pathogenesis (*Sjakste et al., 2016*). Several SNPs and microsatellite alleles localized inside the *PSMA6* proteasome gene and in its vicinity are associated with the risk of T2D (*Sjakste et al., 2007b*, *2007a*). Moreover, *PSMD9* genes SNPs rs74421874, rs3825172 and rs14259 were reported to be associated with DR in T2D and non-diabetic retinopathy in Italians (*Gragnoli, 2011a*; *Thiruchelvi & Raghunathan, 2021*), as well as *PSMD9* SNPs were linked with other microvascular T2D complications-neuropathy (*Gragnoli, 2011b*), neuropathy (*Gragnoli, 2011c*) and late-onset T2D itself (*Gragnoli, 2010a*). *PSMD9* association was also observed with maturity-onset diabetes of the young type 3 (MODY3) (*Gragnoli, 2010b*).

Two *PSMB8* SNPs, rs3763365 and rs9276810, were also genetic risk factors for T1D development (*Zaiss et al., 2011*). It is observed that *PSMB8*-B/B may be the protective genotype, but *PSMB8*-B/A could be a susceptible genotype for T1D growth in the Asian population (*Ding et al., 2001*). Another study concluded that allelic and dominant models of *PSMB8* G37360T could be protective in T1D in the Caucasian population. Still, dominant model of *PSMB9 CfoI* could be a risk factor for T1D in the Asian population (*Xu et al., 2020*).

Genetic deletion of proteasome activator genes (PA28α and PA28β) protected the diabetic mice against renal and retinal microvascular injury compared with wild-type diabetic mice. The authors concluded that hyperglycemia promoted PA28-mediated alteration of proteasome activity in vulnerable perivascular cells resulting in expression of the pro-inflammatory proteins osteopontin and MCP-1, microvascular injury and development of diabetic nephropathy and DR. Thus, decrease in the proteasome activation

**Table 1 UPS-associated genetic factors, diabetes, and its complications.**

| Gene description | Gene name | SNPs | Type of DM | Association observed | Population |
|---|---|---|---|---|---|
| Proteasome 20S Subunit Alpha 3 | *PSMA3* | rs2348071 | T1D (*Sjakste et al., 2016*) | susceptibility to T1D | Latvian (*Sjakste et al., 2016*) |
| Proteasome 20S Subunit Alpha 6 | *PSMA6* | rs1048990 rs2277460 | T1D (*Sjakste et al., 2016*) T2D (*Sjakste et al., 2007b; Sjakste et al., 2007a; Barbieri et al., 2008; Kim et al., 2008*) | susceptibility to T1D and T2D | Latvian (*Sjakste et al., 2016; Sjakste et al., 2007b, 2007a*), Finnish (*Sjakste et al., 2007a*), Korean (*Kim et al., 2008*), Caucasian (*Sjakste et al., 2007b; Sjakste et al., 2007a; Barbieri et al., 2008*) |
| Proteasome 26S Subunit, ATPase 6 | *PSMC6* | rs2295826 rs2295827 | T1D (*Sjakste et al., 2016*) | susceptibility to T1D | Latvian (*Sjakste et al., 2016*) |
| Proteasome 26S Subunit, Non-ATPase 9 | *PSMD9* | rs74421874 rs3825172 rs14259 | T2D (*Gragnoli, 2011a; Gragnoli, 2011b; Gragnoli, 2011c; Gragnoli, 2010a*) MODY3 (*Gragnoli, 2010b*) | susceptibility to T2D and diabetic retinopathy, diabetic nephropathy,diabetic neuropathy in T2D; susceptibility to MODY3 | Italian (*Gragnoli, 2011a, 2011b, 2011c; Gragnoli, 2010a, 2010b*) |
| Proteasome 20S Subunit Beta 8 | *PSMB8* | rs3763365 rs9276810 | T1D (*Zaiss et al., 2011; Ding et al., 2001; Xu et al., 2020*) | susceptibility to T1D | Caucasian (*Zaiss et al., 2011*), Asian (*Ding et al., 2001; Xu et al., 2020*) |
| | | G37360T | T1D (*Xu et al., 2020*) | protective in T1D | Caucasian (*Xu et al., 2020*) |
| Proteasome 20S Subunit Beta 9 | *PSMB9* | The dominant model of CfoI | T1D (*Xu et al., 2020*) | susceptibility to T1D | Asian (*Xu et al., 2020*) |
| Proteasome 20S Subunit Beta 5 | *PSMB5* | rs2230087 | T2D (*Kim et al., 2008*) | susceptibility to T2D | Korean (*Kim et al., 2008*) |

**Note:**
MODY3, maturity onset diabetes of the young type 3; T1D, type 1 diabetes; T2D, type 2 diabetes.

by PA28 appeared to be favorable for protection against the development of the DR (*Yadranji Aghdam & Mahmoudpour, 2016*).

Genetic factors of the UPS associated with diabetes and its complications are summarized in Table 1.

## UPS, diabetes-induced oxidative stress and DR

Diabetes is a state of chronic oxidative stress and hypoxia induced by hyperglycemia (*Brownlee, 2005*). 26S proteasomes are susceptible to oxidative stress, probably due to the oxidation of essential amino acids in the proteasome activator PA700, which mediates the ATP-dependent proteolysis of the 26S proteasome. In contrast 20S proteasome-meditated degradation is much more resistant, even in the presence of high concentrations of hydrogen peroxide ($H_2O_2$) (*Reinheckel et al., 2000*). Continuous oxidative stress elevates the amount of damaged proteins and UPS impairment which leads to their build-up in cells (*Dong et al., 2004; Menéndez-Benito et al., 2005*).

Diabetes-induced oxidative stress plays a vital role in the pathogenesis of DR (*Fernandes et al., 2014; Sasaki et al., 2010; Kowluru & Chan, 2007*). Levels of reactive oxygen and

nitrogen species, including the highly reactive oxidant peroxynitrite are increased in diabetic retinas (*Zhang et al., 2002*). *Fernandes et al. (2014)* reported that increased oxidative stress in diabetic retinas led to the inactivation of the 20S proteasome in Goto-kakizaki rats with dyslipidemia and accumulation of ubiquitinated proteins that affected the chymotrypsin-like activity of the proteasomes. The application of atorvastatin (a synthetic lipid-lowering agent) had a local antioxidative effect that restored the ubiquitin-proteasome pathway in an atherogenic diet-fed rats (*Fernandes et al., 2014*). In this case decrease in proteasomal activity appears to be unfavorable.

Transcription factor NF-E2 related factor 2 (NRF2) is one of the stress-response proteins for the antioxidative defense of the cell (*Chapple, Siow & Mann, 2012*). Under unstressed conditions, Kelch-like ECH-associated protein 1-nuclear factor (KEAP1) serves as an adaptor for ubiquitin E3 ligase and promotes proteasomal degradation of NRF2. NRF2 is stabilized when KEAP1 is inactivated under oxidative/electrophilic stress conditions. In this case, NRF2 promotes expression of cytoprotective genes. NRF2 binding to KEAP1 increases in diabetes, leading to its proteasomal degradation and decreased cell-stress response. In DR, epigenetic changes of *KEAP1* gene can lead to decreased NRF2 expression and impaired antioxidative response (*Mishra, Zhong & Kowluru, 2014*). Also, NRF2 function in diabetes is suppressed by regulated in development and DNA damage responses 1 (REDD1). Specifically, REDD1 suppressed NRF2 stability by promoting its proteasomal degradation independently of NRF2's interaction with KEAP1, preventing antioxidative response in retinal cells of diabetic mice (*Miller et al., 2020*). These findings suggest that targeting proteasomal degradation of NRF2 is a promising approach in DR, as increased proteasomal degradation of NRF2 seems to be harmful.

## UPS, hypoxia, and DR

Diabetes is a state of chronic hypoxia due to the glycation of haemoglobin and increased oxidative stress (*Takiyama & Haneda, 2014*). Proteasome activity is impaired in response to hyperglycemia-associated hypoxia (*Aghdam et al., 2013*; *Miyata, Suzuki & van Ypersele de Strihou, 2013*). Hypoxia also alters the substrate specificity, of proteasomes. During the hypoxia-triggered decomposition of the 26 proteasomes, the role of 20S complexes increases. The latter can bind non-ubiquitinated proteins and degrade ubiquitin itself. 20S proteasome recognizes misfolded proteins and proteins with exposed Cys residues. Products of proteolysis by 20S proteasome are abundant in hypoxic cells and human hearts after heart failure (*Sahu & Glickman, 2021*; *Sahu et al., 2021*).

A protein important for the pathogenesis of DR is hypoxia-induced factor 1 alpha (HIF1-alpha). HIF-alpha went through hydroxylation by prolyl hydroxylase domain in a normoxic conditions, resulting in proteasomal degradation. In hypoxic conditions HIF1-alpha is not hydroxylated, but is stabilized in cytosol and forms a heterodimer with HIF1-beta, which binds to hypoxia-responsive elements in the nucleus and activates downstream genes such as GLUT1, erythropoietin, vascular endothelial growth factor (VEGF) (*Hirakawa, Tanaka & Nangaku, 2017*) and angiopoietin 2 (*Hussain et al., 2019*) involved in pathogenic angiogenesis in DR.

## Impact of hyperglycemia on E3 ligases in the retina

E3 ligases in mammals are commonly grouped into three classes: really interesting new genes (RINGs), homologous to E6AP C terminus (HECTs), and RING-between-RINGs (RBRs). The E3 ligases belonging to any of the classes catalyze covalent attachment of ubiquitin to a Lys residue in the target protein. However, they differ in structure and mechanisms of action (*Zheng & Shabek, 2017*). In early reports, six E3 ligases of the RING family (TOPORS, UBR1, TRIM2, PARKIN, SIAH1, and MDM2) and two HECTs (HERC6 and NEDD4) were detected in the retinas of mammals (*Campello et al., 2013*). Later it was reported that representatives of the TRIM family were numerous in the retina, TRIM9 was revealed (*Chowdhury et al., 2018*). TOPORS, a dual E3 ubiquitin, and SUMO1 ligase, are essential for retinal homeostasis. The enzyme interacts with 26S protease regulatory subunit PSMC1 (*Czub et al., 2016*). It was shown SIAH family of E3 ubiquitin ligases play a role in optic fissure fusion and identified CDHR1A and NLZ2 as potential targets of SIAH (*Piedade et al., 2020*; *Pereira Piedade, Veith & Famulski, 2019*).

Both ubiquitin and E3 ligases are upregulated in human retinal endothelial cells exposed to high glucose concentration (*Luo et al., 2015*). On the contrary, in diabetic rats, SIAH1 expression was decreased in 2-month diabetic rats, but no significant change was observed in 4-month diabetic rats compared with their controls (*Shruthi, Reddy & Reddy, 2017*). SIAH1 is involved in the accelerated degradation of synaptophysin—a major synaptic vesicle protein—in diabetic mice (*Ozawa et al., 2011b*).

E3 ligase PARKIN plays a unique role in the retina cells, as it is involved in the mitophagy process. In damaged mitochondria, the outer membrane is depolarized. Serine/threonine kinase PINK1 (PTEN-induced putative kinase 1) serves as a sensor for the mitochondrial polarization state. In physiologic conditions, the polarized mitochondrial PINK1 is imported into the mitochondria and degraded by the protease PARL (presenilin associated rhomboid-like protein) and proteasomes. Mitochondrial damage results in the accumulation of PINK1 on the outer mitochondrial membrane and recruitment of PARKIN from the cytosol. PINK1 phosphorylates ubiquitin and the ubiquitin-like domain of PARKIN. When activated, the activity of PARKIN E3 ligase ubiquitinates numerous downstream autophagosome-related proteins: mitofusins MFN1 and MFN2, fission protein FIS, its adaptor TBC1D15 and translocase TOMM20 and TOMM70 that facilitate movement of proteins across the outer mitochondrial membrane. In this way PARKIN stimulates the local formation of autophagosomes (*Devi et al., 2017*; *Huang et al., 2018*; *Zhou et al., 2019*; *Zhou et al., 2020*). Treatment of the retina-derived cells with high glucose upregulates the PINK1/PARKIN pathway (*Devi et al., 2017*; *Huang et al., 2018*; *Zhou et al., 2019*; *Zhou et al., 2020*), although some authors report down-regulation of the PINK1/ PARKIN pathway in hyperglycemia (*Zhang et al., 2019*). The path is considered to play a protective role against hyperglycemia (*Huang et al., 2018*). This pathway is a molecular target for possible remedies against DR; however different compounds produce opposite effects: the incretin glucagon-like peptide-1 inhibits mitophagy *via* PINK1/ PARKIN (*Zhou et al., 2020*), but a saponin notoginsenoside stimulates PINK1/PARKIN -mediated mitophagy (*Zhou et al., 2019*).

## UPS and diabetes-induced ER stress in the retina

Downregulation of ERAD components was documented in experimental diabetes (*Yang et al., 2015*). *Shruthi et al. (2019)* observed changes in ERAD components in the cerebral cortex of animals with experimental diabetes. Upregulation of ERAD components (HRD1, Derlin1, and VCP) in early diabetes is observed and might represent a defensive mechanism against ER stress. However, continuing chronic hyperglycemia and oxidative stress leads to a significant decrease of the mentioned ERAD components, further elevating ER stress (*Shruthi et al., 2019*).

ER stress is also involved in the development of DR (*Li et al., 2009*; *Shruthi, Reddy & Reddy, 2017*), possibly because of reduced amounts of E1 and HRD1 (ER stress-induced protein with ubiquitin ligase-like activity), components of UPS. Treatment with a chemical chaperone 4-phenylbutyric acid (4-PBA) altered retinal cells, restored deubiquitinases and improved ER stress-related cell death (*Shruthi, Reddy & Reddy, 2017*). In cultured human retinal pericytes that are exposed to high glucose treatment, the induction of ER stress was associated with the upregulation of proteasome activator 11S REG (PA28 a/-β) (*Aghdam et al., 2013*; *Aghdam & Sheibani, 2013*; *Zhong, Wang & Zhang, 2012*).

## UPS and other diabetes-induced pathogenic pathways in DR

UPS interferes with several pathogenetic pathways developing in hyperglycemia conditions, including polyol pathway flux, activation of protein kinase C isoforms, increased hexosamine pathway flux, and advanced glycation end-product formation (*Safi et al., 2014*). Methylglyoxal formed in the conditions of hyperglycemia modifies several proteasome proteins and decreases chymotrypsin-like proteolytic activity of proteasomes (*Queisser et al., 2010*). Proteolytic pathways are the last line of defense against advanced glycation end products (AGE)-derived proteotoxicity. Different proteolytic pathways (UPS and autophagy) act to avoid the accumulation of toxic AGEs (*Aragonès et al., 2020*).

Conversely, protein kinase Cβ (PKCβ) is a serine-threonine kinase associated with obesity and diabetic complications; its activation contributes to weight gain. PKCβ positively regulates fat mass and obesity-associated protein (FTO), an RNA demethylase involved in the development of obesity. The over expression of PKCβ suppresses ubiquitin-proteasome degradation of FTO, whereas PKCβ inactivation enhances FTO degradation (*Tai et al., 2017*).

The hyperglycemia-induced oxidative stress and the following apoptosis activates the phosphatase and tensin homolog (PTEN) signaling cascade. PTEN is regulated by several E3 ligases (NEDD4-1, XIAP, and WWP2), which trigger its degradation *via* UPS. In hyperglycemia, a chaperone carboxyl terminus of Hsc70 interacting protein (CHIP) with E3 ligase activity maintains PTEN expression preventing its deleterious effects (*Ali et al., 2021*).

## UPS and neurodegeneration in DR

Angiotensin II and its receptors angiotensin II type 1 receptor (AT1R) and type 2 receptor (AT2R) become upregulated in experimental diabetic eye disease (*Ozawa et al., 2011a*; *Kurihara et al., 2008*). Synaptophysin is a major synaptic vesicle protein that is

co-expressed with AT1R in the inner layers of the retina (*Kurihara et al., 2006*). Synaptophysin levels are reduced in neurodegenerative diseases such as dementia, Parkinson's disease, and Alzheimer's disease (*Zhan, Beyreuther & Schmitt, 1993*; *Sun et al., 2004*). In diabetes, angiotensin II and AT1R together with AT1R's downstream extracellular signal are upregulated-related protein kinase (ERK) activation in retina (*Kurihara et al., 2008*), that induces synaptophysin degradation. Therefore, activating the angiotensin II-AT1R-ERK pathway increases the ubiquitin-conjugated synaptophysin protein levels (*Kurihara et al., 2008*), leading to decreased synaptophysin levels in experimental DR (*Ozawa et al., 2011a*, *2011b*). Increased proteasomal degradation of synaptophysin compromises synaptic activity, worsens neuronal cell survival and vision in diabetes. However, synaptophysin degradation can be inhibited by blocking AT1R signaling *in vivo* by angiotensin receptor blockers. Telmisartan and valsartan significantly reversed the diabetes-induced changes in the electroretinogram, suggesting that the suppression of diabetes-induced retinal dysfunction and synaptophysin degradation is a class effect for angiotensin receptor blockers (*Ozawa et al., 2011b*). Antioxidant lutein can also prevent ERK activation and the following reduction of synaptophysin in the diabetic retina (*Sasaki et al., 2010*).

Decreased rhodopsin levels have been observed in rats with experimental diabetes and may be associated with vision impairment in early diabetes (*Malechka et al., 2017*). Degradation of rhodopsin, an essential protein for photoreceptor function, is mediated by a STAT3-dependent E3 ubiquitin ligase, Ubr1, up-regulated in inflamed retinas (*Ozawa et al., 2008*), suggesting impairment of UPS regulation as one of the reasons for the decrease of rhodopsin in the diabetic retina.

Reduced protein expression of UPS components was observed in retinal ganglion and horizontal cells (*Shruthi, Reddy & Reddy, 2017*; *Bonfanti et al., 1992*) possibly contributing to neurodegeneration in diabetic eye disease (*Sasaki et al., 2010*; *Ristic, Tsou & Todi, 2014*).

A variety of novel preparations targeting different components of UPS are under development and testing in neurodegenerative diseases (*Thibaudeau & Smith, 2019*). As neurodegeneration is a crucial mechanism of diabetic eye disease, we are looking forward to preclinical studies in DR.

## UPS in the adaptive mechanisms in DR

Hyperglycemia is associated with the increased ubiquitination and proteasomal degradation of some proteins, that might represent an adaptive mechanism (*Fernandes et al., 2004*; *Fernandes, Hosoya & Pereira, 2011*). In DR, in the setting of oxidative stress, subcellular redistribution of glucose transporter 1 (GLUT1) occurs (*Fernandes, Hosoya & Pereira, 2011*), which is the main isoform of glucose transporters in retinal endothelial cells (*Wieman, Wofford & Rathmell, 2007*). In conditions of increased oxidative stress, endothelial cells upregulate ubiquitin-proteasome pathway with subsequent increased turnover of ubiquitin conjugates. GLUT1 seems to be a monoubiquitinated or diubiquinated and targeted accordingly for lysosomal degradation, decreasing glucose transport into retinal endothelial cells as well as the associated glycotoxicity (*Fernandes, Hosoya & Pereira, 2011*).

**Table 2 Ubiquitin-proteasome system-affecting drugs and remedies.**

| Drug | Observed effects |
|---|---|
| Atorvastatin | Reduces the levels of oxidative stress induced by the atherogenic diet and restores proteasome activity in the diabetic Goto-kakizaki rats (*Fernandes et al., 2014*). |
| Glucagon-like peptide-1 | Inhibits mitophagy *via* PINK1/PARKIN (cytosolic E3-ubiquitin ligase) pathway in retinal ganglion cells *in vitro* and diabetic rats (*Zhou et al., 2020*). |
| Saponin Notoginsenoside R1 | Enhances mitophagy and suppresses oxidative stress inflammation by activating PINK1/PARKIN (cytosolic E3-ubiquitin ligase) pathway in the retina of db/db mice (*Zhou et al., 2019*). |
| Chemical chaperone 4-phenylbutyric acid (4-PBA) | Restores the levels of deubiquitinases and improves ER stress-related cell death in the retinas of diabetic rats (*Shruthi, Reddy & Reddy, 2017*). |
| Angiotensin receptor blockers – telmisartan and valsartan | Protect against ubiquitination and degradation of synaptophysin in the retina of mice streptozotocin diabetes model (*Kurihara et al., 2008*). |
| Antioxidant lutein | Prevents ROS generation and synaptophysin degradation in the retina of murine diabetes models (*Sasaki et al., 2010*). |
| Trichostatin A | Enhances ubiquitination of p300 – histone acetyltransferase leading to reduced levels of NADPH oxidase 4 (Nox4), a mediator of angiogenesis, and inhibits angiogenesis *in vitro* (*Hakami, Dusting & Peshavariya, 2016*). |
| Proteasome inhibitor MG132 | This leads to inhibition of TGF-beta activation affects the NRF2 pathway and antioxidative capacity in *in-vitro* and *in-vivo* models of diabetes (*Huang et al., 2014*; *Kong et al., 2017*; *Luo et al., 2011*; *Gao et al., 2014*; *Gao et al., 2013*). |
| Inhibitors of heat shock protein 90 (Hsp90): geldanamycin, its analogs, and deguelin | Promotes proteasomal degradation of HIF1-alpha modulating hypoxia-induced pathways of retinal neovascularization (*Vadlapatla, Vadlapudi & Mitra, 2013*). |

**Note:**
4-PBA, chemical chaperone 4-phenylbutyric acid; E3, ubiquitin ligase; ER, endoplasmic reticulum; HIF1-alpha, hypoxia – induced factor 1 alpha; Hsp90, heat shock protein 90; Nox4, nicotinamide adenine dinucleotide phosphate oxidase 4; NRF2, nuclear factor-erythroid factor 2-related factor 2; PINK1, phosphatase and tensin homologue-induced putative kinase 1; ROS, reactive oxygen species; TGF-beta, transforming growth factor beta.

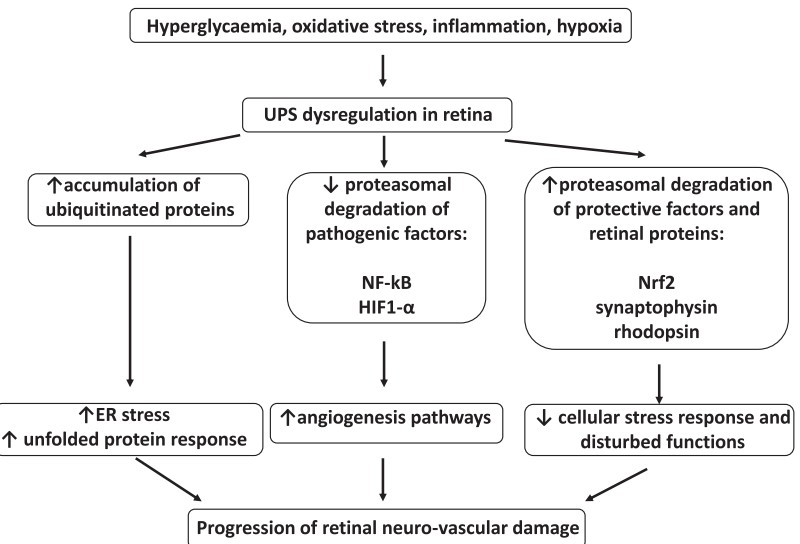

**Figure 1 Involvement of ubiquitin-proteasome system in the pathogenesis of diabetic retinopathy.** UPS, ubiquitin–proteasome system; NF-kB, nuclear factor kB; HIF1-α, hypoxia-inducible factor 1α; Nrf2, nuclear factor erythroid 2–related factor 2; ER, endoplasmic reticulum

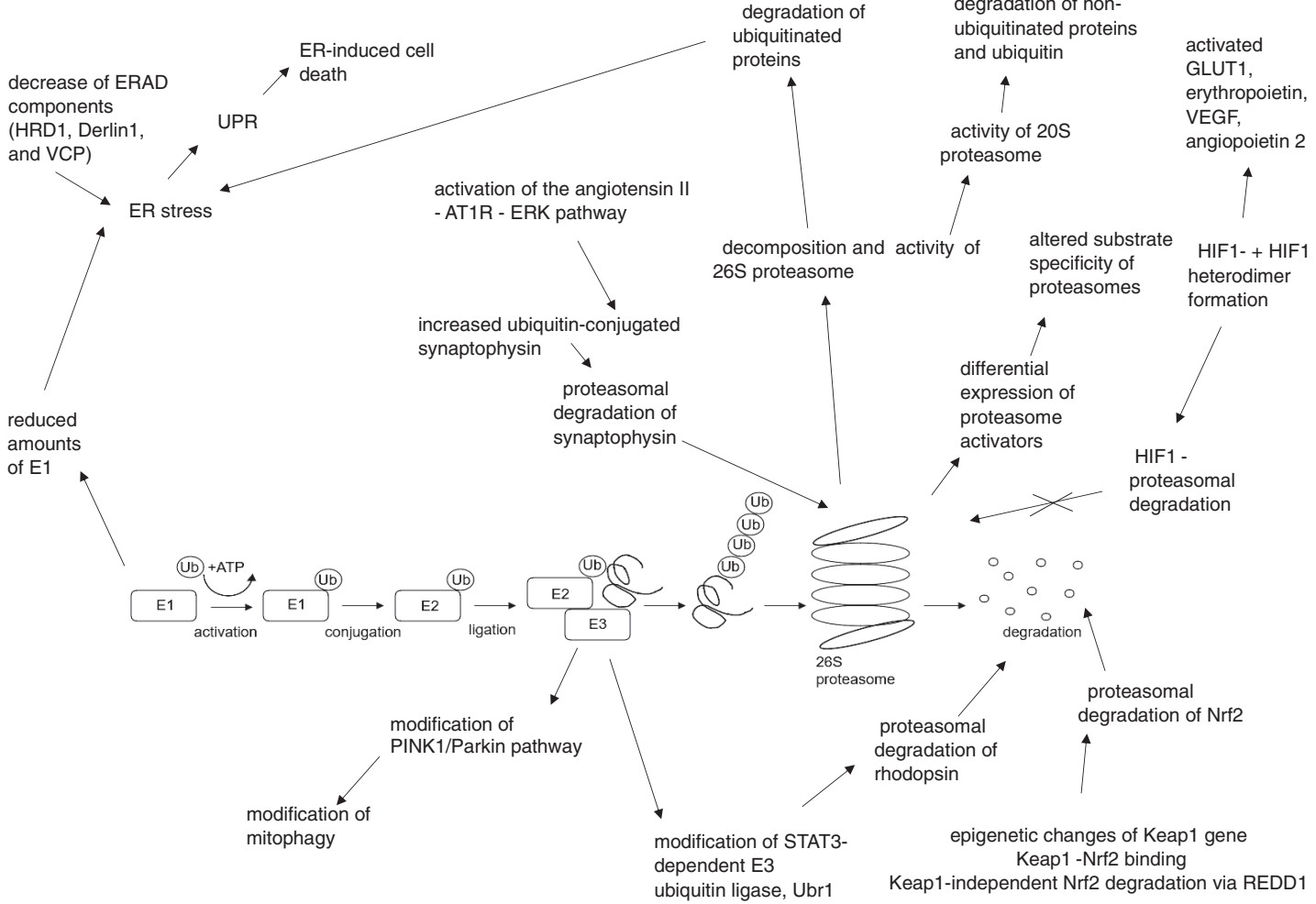

**Figure 2 Hyperglycemia, hypoxia and oxidative stress-driven modifications of ubiquitin-proteasome system (UPS) in diabetes and diabetic retinopathy.** AT1R, angiotensin II type 1 receptor; Derlin1, degradation in endoplasmic reticulum protein 1; E1, ubiquitin activating enzyme; E2, ubiquitin binding enzyme; E3, ubiquitin ligase; ER, endoplasmic reticulum; ERAD, endoplasmic reticulum – associated protein degradation; ERK, extracellular signal – related protein kinase; GLUT1, glucose transporter 1; HIF1-α, hypoxia – induced factor 1 alpha; HIF1β, hypoxia – induced factor 1 beta; HRD1, ERAD-associated E3 ubiquitin-protein ligase HRD1; Keap1, kelch-like ECH-associated protein 1-nuclear factor; Nrf2, nuclear factor-erythroid factor 2-related factor 2; PINK1, phosphatase and tensin homologue-induced putative kinase 1; REDD1, regulated in development and DNA damage response – 1; STAT3, signal transducer and activator of transcription 3; Ub, ubiquitin; Ubr1, E3 ubiquitin-protein ligase UBR1; UPR, unfolded protein response; VCP, valosin-containing protein; VEGF, vascular endothelial growth factor.

## UPS targeted therapies in diabetic microangiopathy

The summary of UPS-affecting drugs and remedies is given in Table 2.

Proteasome inhibitors have entered clinical practice to treat malignancies, especially multiple myeloma. Current data on UPS-affecting treatments in diabetic microangiopathy is very limited. A compound under investigation Trichostatin A induced ubiquitination of p300-histone acetyltransferase leading to reduced levels of NADPH oxidase 4 (NOX4), a mediator of angiogenesis, and an inhibited angiogenesis in an *in vitro* model (*Hakami, Dusting & Peshavariya, 2016*). There are slightly more data on UPS-affecting treatments in diabetic nephropathy. Proteasome inhibitor MG132 inhibits TGF-beta activation, and

affects NRF2 pathway and antioxidative capacity, all involved in the pathogenesis of microvascular disease in diabetes (*Huang et al., 2014*; *Kong et al., 2017*; *Luo et al., 2011*; *Gao et al., 2014*; *Gao et al., 2013*). Furthermore, inhibitors of heat shock protein 90 (HSP90) which stabilizes HIF1-alpha, can promote proteasomal degradation of HIF1-alpha modulating hypoxia-induced pathways of retinal neovascularization. Examples of HSP90 inhibitors include geldanamycin, its analogs and deguelin, which demonstrated promising results in experimental studies (*Vadlapatla, Vadlapudi & Mitra, 2013*).

The summary of UPS involvement in the pathogenesis of DR is shown in Figs. 1 and 2.

## CONCLUSIONS

In diabetes, the overall activity of the UPS is impaired by oxidative stress, hyperglycemia, and hypoxia. The binding of proteasome activator 19S to 20S proteasome is inhibited, leading to slower degradation of polyubiquitylated substrates. Downregulation of some UPS components is associated with ER stress and over-activation of UPR, eventually leading to retinal cell death. Hypoxia-induced decreased proteasomal degradation of HIF1-alpha leads to pathological angiogenesis *via* VEGF, angiopoietin 2, and GLUT1 pathways. At the same time, a selective increase of the proteasomal degradation of individual proteins is possible in the retina in diabetic conditions and contributes to DR. For example, cell stress response mediator NRF2 is excessively degraded due to its increased binding to KEAP1 in diabetes. Proteasomal degradation of synaptophysin is a major synaptic vesicle protein in retina, it is increased due to upregulation of the angiotensin II receptors in diabetes, and contributes to neurodegeneration. Dysregulation of the UPS also leads to rhodopsin degradation. Both stimulators and inhibitors of the UPS-mediated degradation of individual proteins are tested as remedies against the DR.

An obvious problem in the area is the small number of human studies. To improve the understanding of UPS regulation in diabetic eye disease and to promote the development of novel therapies, future directions for the research might include: validation of genetic association studies in different populations of T1D and T2D patients with larger sample size, development, and validation of easily assessable biomarkers of the UPS activity in patients with different severity of DR in body fluids (tears, blood, urine, *etc.*), search for possibilities of tissue and substrate-specific stimulation and inhibition of proteasomal degradation in *in vitro* and *in vivo* experimental studies, clinical studies on intra-vitreal UPS modulators in progressive DR.

### Funding

The work was supported by the project of the Mutual funds Taiwan-Latvia-Lithuania "Novel biomarkers of diabetic retinopathy: epigenetic modifications of genes of ubiquitin-proteasome system, telomere length and proteasome concentration" and a fundamental research grant in Biomedicine and Pharmacy "Research of biomarkers and natural substances for acute and chronic diseases' diagnostics and personalized treatment"

by the Faculty of Medicine, University of Latvia. The funders had no role in study design, data collection and analysis, decision to publish, or preparation of the manuscript.

## Grant Disclosures
The following grant information was disclosed by the authors:
Mutual Funds Taiwan-Latvia-Lithuania.
Biomedicine and Pharmacy, Faculty of Medicine, University of Latvia.

## Competing Interests
Zane Svikle, Beate Peterfelde, Nikolajs Sjakste, Kristine Baumane, Rasa Verkauskiene, and Chi-Juei Jeng report no conflicts of interest. Jelizaveta Sokolovska reports lecture fees and educational grants from Sandoz, Sanofi, MSD, NovoNordisc, AstraZeneca, Grindex outside the submitted work. Nikolajs Sjakste is an Academic Editor for PeerJ.

## Author Contributions
- Zane Svikle conceived and designed the experiments, performed the experiments, analyzed the data, prepared figures and/or tables, and approved the final draft.
- Beate Peterfelde performed the experiments, analyzed the data, authored or reviewed drafts of the article, and approved the final draft.
- Nikolajs Sjakste conceived and designed the experiments, authored or reviewed drafts of the article, and approved the final draft.
- Kristine Baumane analyzed the data, authored or reviewed drafts of the article, and approved the final draft.
- Rasa Verkauskiene analyzed the data, authored or reviewed drafts of the article, and approved the final draft.
- Chi-Juei Jeng analyzed the data, authored or reviewed drafts of the article, and approved the final draft.
- Jelizaveta Sokolovska conceived and designed the experiments, prepared figures and/or tables, and approved the final draft.

## Data Availability
This is a literature review, there are no raw data or codes.

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
