# Peer review of "Ubiquitin-proteasome system in diabetic retinopathy"

_PeerJ, doi:10.7717/peerj.13715_

## Round 0.1 · original submission · Major Revisions

As you will see from the reviewers comments, there is something of a split decision here. Having read the review myself, I am minded to give you an opportunity to address the substantive concerns raised, in particular by reviewer-3 (but please address all comments).

However, it is clear that the article needs a major re-focus and re-write. Many of the issues raised arise as a consequence of language. Please could you strive for more clarity - in places the manuscript is confusing as the use of English isn't ideal.

I hope you will find these comments helpful, and ask that you address all points in your rebuttal.

·

Basic reporting

The English language could be improved to ensure maximum clarity. Sufficient background is provided. Figures need to be added for further clarity and ease of understanding.

Experimental design

The survey methodology is nice. PubMed and Medline databases are used in search of the scientific literature. It would be nice to use the Google Scholar search engine too for any missing literature in this scientific area.

Validity of the findings

1. In the abstract (line no. 29) of the manuscript, the authors stated that ‘In diabetes, the overall activity of the UPS is inhibited by the oxidative stress’. This is true in major conditions of diabetes, however, in diabetic myopathy, the over activation of UPS leads to muscle wasting. Hence, authors should consider this aspect too.
2. The E3 ligases are the vital components of UPS that confers specificity to UPS. Hence, authors need to include the latest developments in identifying retinal E3s and their role in diabetes and diabetic retinopathy.
3. PubMed and Medline databases are used in search of the scientific literature. It would be nice to use the Google Scholar search engine too for any missing literature in this scientific area.
4. Authors need to follow the general rules of writing gene names (Line no. 179, 185, 186, 188, 189, 190 etc.).
5. It is very informative that the authors depicted the role of UPS in DR as Figure 1. However, the manuscript has only one illustration and the addition of a few more adds to the ease of understanding.
6. Conclusion is well written and It could be better if the authors add ‘Future Directions’ of this area
7. Few typos are identified, for example:
Ubiquitin enzymes instead of ubiquitinating or ubiquitination enzymes in line no. 96.
Ubiquinate instead of ubiquitinate in line no. 135
Ubiquitine instead of ubiquitin in Figure 1 legend
Hiperglycaemia instead of hyperglycaemia in the Figure
8. The authors abbreviated diabetic retinopathy as DR. However, it is not followed consistently.
9. The English language could be improved to ensure maximum clarity. Some examples where the language could be improved include lines 36, 53, 141, 307, etc.

Additional comments

No comments

·

Basic reporting

no comment

Experimental design

no comment

Validity of the findings

Several sentence need to be reworded to make them valid. The specific examples are given in the additional comments.

Additional comments

The authors present introductory level review of the role of the UPS in DR. Mostly, the review was comprehensible, though the authors are strongly encouraged to revise selected lines (listed below). Additionally, the authors are also strongly encouraged to consult a colleague proficient in English, and familiar with the subject matter, or contact a professional editing service to make the article more readable and clear.
There are few factual errors that the authors should revise prior to acceptance (see below). Generally, the review is adequately written and cited, save the issues listed below. The references consulted were appropriate, indicating an appropriate methodology of survey, but organization of study comparisons can be better executed. In the end the review provides an introductory view of the UPS and its involvement in physiology (e.g. oxidative stress) that occurs in DR, that would be of interest to someone who is unfamiliar with these topics.

Line 29 – 31: The two sentences do not seem to flow together, possibly due to the usage of “however”.
Line 58: “microangiopathy” should be defined
Line 111: not clear what is meant by “density “ of ub substrate? Authors should clarify.
Line 111-112: The fate of ubiquitinated substrates does not only depend on the number of ubiquitin, but also the geometry of ubiquitin chains. (French et. al, 2021, Cell Discovery)
Line 112 should be e.g. instead of i.e. as many different chain lengths can target for degradation as well, even multiple monoubiquitinations.
Line 115 – 117: There is one constitutive proteasome, which is the catalytic core also referred to as the 20S. The others such as the 19S regulator (which associates with the 20S to form the 26S; the 19S is not a proteasome itself) or the 11S regulators (PA28αβ, PA26) are collectively referred to as Proteasome Activators, regulatory complexes of the proteasome. (Stadtmueller & Hill, 2012, Molecular Cell)
Line 118: The appropriate abbreviation for the immunoproteasome is 20Si, as referred to throughout the proteasome field.
Line 122 – 124: The immunoproteasome is assembled using a slightly different group of subunits from the constitutive proteasome, hence the difference in proteolysis activity (ie. the reduction of caspase-like activity), this should be clarified. e.g. “misses caspase activity” is not clear.
Line 125 “compensate the ubiquitin process” is not clear.
134 is wrong, ERAD proteins are NOT degraded in the ER lumen. Instead proteins are extracted from the ER and degraded in the cytosol. This could be an English problem but must be fixed, because it is false as stated.
Line 144, use of the word “derangements” is awkward and non-tradition and unclear what is meant.
Line 170, Protein degradation of what?
Line 203, deletion of pa28 may not result in a decrease in proteasome activity, this should be worded more carefully. E.g. PA28 binding to the 20S could prevent 26S formation decreasing other types of proteasome activity. Also the proteasome “activity” can be assessed via many methods, the type of activity that is being referred to should be made clear.
Line 205 – 208: The 26S proteasome is known to disassemble in the state of hypoxia. It is not that the 26S is ineffective at degrading oxidized proteins, but that the hypoxic environment simply decreases the ratio of 26S to 20S. The references utilized for this point are appropriate though the interpretation is not accurate.
Line 203 & 224: As noted by the authors, several studies contradict as to whether the increase or decrease in proteasome activity is involved in the progression of diabetic retinopathy. The review can be improved by discussing substrate-specificity of the 20S when associated with different regulatory complexes. Perhaps the point is not to simply stimulate or inhibit proteasome activity but to target specific proteasome substrates.

Reviewer 3 ·

Basic reporting

In this manuscript, Svikle et al. tried to summarize the biology of UPS associated to DR. The UPS biology has not reviewed recently in the context of DR and it could be interest for the journal’s readers. Unfortunately, the manuscript is not well-written and a high % of citations are other reviews (not original papers). Regarding DR, it not covered if UPS activity is different in the different retinal cellular components or how diabetes leads to UPS dysregulation in retina. The manuscript seems to be an immature draft (see below) and it should be re-written from the scratch. It is absolutely incredible that 7 authors did not detect so many errors in the text. It does not fulfil minimal requirement to be published in the journal.


1. The abstract of the manuscript is chaotic. it should provide an overview to the readers without making them read the entire paper.
2. The most important issue is that it is not clear if the overall activity of UPS is increased or decreased in DR. Would enhancers (or inhibitors) of UPS have benefits to combat DR? How is UPS activity by oxidative stress in DR? Once the reader finishes the reading….. there is not a clear message.
3. Why were the terms “telomeres” and “telomere length” included in the literature search?? There is nothing about telomeres in the manuscript at all.
4. References 15-24 are missing. No text in the manuscript include those citations. It sounds that a section was completely erased.
5. The sentence “accumulating evidence demonstrates the potential role of UPS in diabetes” is repeated several times (abstract line 2, line 71, line 146). What does it exactly mean?? Do the authors want to say that UPS dysfunction play a major role in pathogenesis of DR??
6. Reference 12 do not support that “regulation (of protein quality) is mainly mediated by the UPS in the retina.
7. The different molecular mechanisms associated with DR are not mentioned (polyol pathway flux, activation of protein kinase C isoforms, increased hexosamine pathway flux, and increased advanced glycation end-product formation (doi: 10.1155/2014/801269). Are those metabolic abnormalities impacting the UPS activity in diabetic retina?? For example, it has been reported that hyperglycemia covalently modifies the 20S proteasome DOI: 10.2337/db08-1565
8. Most of the information in the text is not about diabetic retinopathy. It should be clear when the information is related to retinal biology.

Minor comments:

1. Why is term “novel biomarkers” italicized? (line 67)
2. The first time you use an abbreviation in the text, both the spelled-out version and the short form should be presented. Then, the spelled-out version should not show up anymore (e.g. diabetic retinopathy (line 66, line 248, line 265, etc) . Also, “Unfolded protein response (UPR)” appears twice (line 138 and 153)
3. It's not clear what the authors wanted to say with the sentence “the destiny of ubiquitinated substrates relies on the amount of ubiquitin added” (Line 111).
4. Different font size in line 162.
5. symbols for genes should be italicized whereas symbols for proteins are not italicized and in upper-case.

Experimental design

not applicable

Validity of the findings

not applicable

---

## Round 0.2 · Minor Revisions

Thanks for your efforts at re-focussing the article. Some minor suggestions are made from reviewer-2 which should be easy to address.

·

Basic reporting

The revised manuscript is much better in the language. Sufficient background with references provided.

Experimental design

The sources are adequately cited and the review is organized logically into subsections.

Validity of the findings

no comment

Additional comments

The authors have responded reasonably well to all of my queries.

Reviewer 3 ·

Basic reporting

This reviewer appreciates the effort the authors made to improve the manuscript. The manuscript was refocused and rewritten and, now, the text covers properly the information available about the role of the UPS system in diabetic retinopathy.
Some comments:
• In lines 368-371, the authors include the paragraph “Additionally, a variety of novel preparations targeting different components of UPS are under development and testing in neurodegenerative diseases [19]. As neurodegeneration is a crucial mechanism of diabetic eye disease, we are looking forward to preclinical studies in DR.” in the subsection “UPS targeted therapies in diabetic microangiopathy” This paragraph should be included in the “UPS and neurodegeneration in DR” subsection.

• In lines 104-106, it is not clear what the authors mean by “The fate of ubiquitinated substrate depends on the amount of ubiquitin added A protein with a single ubiquitin molecule attached continues to function in an altered way”. In addition, it should be discussed how UPS degradation is dependent on Lysine chain-specific polyubiquitination (Lys48-linked polyubiquitin chains compared to Lys63 or Lys27, for example).

• It is not clear what the authors mean to say in line 118, in line 214-215, in line 347, lines 211-215, in line 286.

• A reference should be included to support the sentence in lines 172-173.

• The “numerous downstream autophagosome-related proteins” should be listed.

Minor comments:

• The manuscript contains multiple typographic mistakes and some grammatical errors:

o Every time the authors use a term composed of two words (e.g. diabetes-induced), the term appears with hyphens first (–diabetes-induced). The first hyphen must not be included (lines 163, 165, 202, 208, 244, 296, 319,
o A blank space is missing in lines 51, 68, 114, 124, 130, 197, 202, 204 (PA700), 207, 209, 263, 363)
o A blank space should be removed in lines 87, 271, 316, 333.
o A period is missing in line 107 and 205.
o In line 133, “Endoplasmic” should be change to “endoplasmic”. In line 135, “endoplasmic reticulum” should be change to “ER
o In line 138, UPR should be spelled out. The authors use the term UPR throughout the text but it is not defined.
o In line 214, “chymotrysin” should be change to “chymotrypsin activity”.
o In Table1 , “Gene name” should be change to “Gene description” and “Gene Symbol” should be change to “Gene Name”.

o In line 202, the authors use “-hyperglycema-induced”. The term is not correct
o Some words are written without a blank space in between: “neurodegenerationetc” (line40), “assessablemarkers” “severityof” (line 407) “glycotoxicity[116]” (line 263), “action[88]” line (256), “Siah[92, 93]” (line 263), “PSMD9SNP” (line 181), “immunoproteasomeis” (line 120), “subtypesdiffering” (line 115), “PSMB9Cfol” (line189), etc.
o The authors must check the names of all genes and molecules: “pten” (line 347) should be written in capital letters, parkina (Parkin?) and Parkin (lines 257 and 270). Also NRF2 and KEAP1.
o In line 233 the authors use the word “nocive”. This reviewer would suggest the use of “harmful”
o In line 288, the authors include information about REDD1 (regulated in development and DNA damage responses 1), but they refer to the protein as “stress response protein regulated in development and DNA damage 1 (REDD1). It must be corrected
o In line 183 the authors use the term “MODY3”, but it is not defined.
o The authors must check typographic mistakes as the one that appears in line 53: (- .)
o In lines 214 and 215, the authors mention chymotrypsin and atorvastatin, respectively, but the function of these substances is not explained.
o This reviewer suggests to change “In a cultured human retinal pericytes that is exposed to” to “In cultured human retinal pericytes that are exposed to” (line 300)
o in vivo, in vitro and et.al should be written in italics and without a hyphen

Experimental design

N/A

Validity of the findings

N/A

Additional comments

N/A

---

## Round 0.3 · accepted · Accept

Thanks for carefully addressing the outstanding issues raised at the second round of review. I am delighted to accept the paper now.